# A Novel NDT Scanning System Based on Line Array Fast Neutron Detector and D-T Neutron Source

**DOI:** 10.3390/ma15144946

**Published:** 2022-07-15

**Authors:** Sheng Wang, Chao Cao, Wei Yin, Yang Wu, Heyong Huo, Yong Sun, Bin Liu, Xin Yang, Rundong Li, Shilei Zhu, Chunlei Wu, Hang Li, Bin Tang

**Affiliations:** Institute of Nuclear Physics and Chemistry, Mianyang 621900, China; mritn5851@gmail.com (S.W.); ccldyq@gmail.com (C.C.); yinwei-itm@163.com (W.Y.); nuclear-wyang@163.com (Y.W.); huoheyong@163.com (H.H.); sy113sse@163.com (Y.S.); 15528559138m@sina.cn (B.L.); yangx05@126.com (X.Y.); amdom@sohu.com (R.L.); 2016zsl@sina.com (S.Z.); wuchunlei@caep.cn (C.W.); inpcnr@gmail.com (B.T.)

**Keywords:** fast neutron imaging, scanning system, line array detectors, D-T neutron source

## Abstract

A novel non-destructive testing scanning system based on a large-size line array fast neutron detector and compact D-T neutron source has been constructed. The scanning range is up to 1000 mm, and the resolution is better than 1 mm. The fast neutron detection subsystem consists of a polypropylene zinc sulfide scintillator embedded with wavelength-shifting fibers, coupled with a light lens and a scientific CCD camera. With a new rotating tritium target, the lifetime of the compact D-T neutron source could achieve ten hours. The experimental results indicate that the scanning method based on line array fast neutron detector and D-T neutron source is feasible and enables the detection of slits on the order of 0.5 mm in width. Fast neutron tomography has been realized by this detection system too.

## 1. Introduction

Neutron radiography (NR) technology is an important non-destructive testing method. Based on different reaction principles, compared with the X-ray method, NR is more suitable for testing light-material defects covered by heavy-material. NR can be used as a useful supplement to X-ray imaging technology. 

Neutron imaging can use different neutron sources, such as reactor neutron source, spallation neutron source, accelerator neutron source, and isotope neutron source. Among them, the neutron imaging technology based on a reactor or spallation source is relatively mature. Moderated thermal neutrons or cold neutrons are mainly used to carry out testing applications [1,2,3,4,5], and the wide-spectrum neutron beam containing fast neutron components, such as fission neutrons, is used for testing applications too [6,7]. The neutron radiography facility relying on a large neutron source has good beam quality, high imaging collimation ratio, and high beam intensity on the imaging plane, but its disadvantage is that the facilities are large and the cost is high, which cannot meet the user’s local testing needs. 

As for the different neutron energies, thermal or cold neutron radiography has a high resolution, high image contrast, and good image signal-to-noise ratio. However, since the metal materials, such as iron and copper that can be penetrated by thermal neutrons or cold neutrons, are only a few centimeters in size, and the hydrogen materials are centimeter-level, the size of detectable samples of these materials is greatly limited. Compared with thermal neutrons or cold neutrons, fast neutrons have a stronger penetration ability. Especially, fast neutron radiography with energy above 10 MeV has the ability to detect samples with a thickness of tens of centimeters and has developed rapidly in recent years. Due to the difficulty of neutron beam intensity and fast neutron detection, the detection quality of fast neutron radiography cannot be compared with thermal or cold neutron radiography. At present, the research on fast neutron radiography is mainly focused on the compact accelerator neutron source. That is because the compact accelerator neutron source is relatively small, and the cost is not high [8,9,10,11,12]. The detectable field of view (FOV) of fast neutron radiography generally does not exceed 350 × 350 mm^2^ [6,9], so the single detection range is not large. The fluorescent screen coupled CCD camera is used as the main neutron detection method [8,11,12], so the detection efficiency is not high. Therefore, developing a fast neutron radiography detection system with a large detection range and high detection efficiency has a good application prospect.

The neutron spectrum of fast neutron radiography obtained by nuclear reactions, such as D-D or D-Be, is wide [8,9,10]. The fast neutrons of 14 MeV single energy can be obtained by D-T reaction, and the higher neutron yield can be obtained under small deuterium beam intensity. It is suitable for the application of fast neutron radiography [12,13,14]. Therefore, the neutron imaging team at INPC of CAEP has constructed several neutron radiography systems based on a compact accelerator of (D-T) fusion reaction [13,14]. The maximum D-T neutron yield is 1.7 × 10^11^ neutrons/s, and the source size is about 10 mm. Preliminary experiments of fast neutron radiography and tomography were carried out, and holes in the material could be reconstructed clearly [15]. Due to the low detection efficiency and low neutron yield, getting one projection image usually requires several minutes, and one tomography even costs several days [13,15]. Moreover, because of the high beam power intensity, the static tritium target needs to be replaced in several hours. This limits the neutron source lifetime and affects the tomography efficiency and reconstruction quality.

In order to enlarge the applications of fast neutron radiography, a new fast neutron detector with higher detection efficiency and larger FOV is necessary. The D-T neutron source needs to be improved too. After learning from the X-ray scanning testing method [16,17], a new fast neutron scanning imaging detecting system was designed. This system can provide the largest detection range among the traditional fast neutron radiography systems. Based on a former study on polypropylene zinc sulfide scintillator embedded with wavelength-shifting fibers [18,19,20], a large-size line array fast neutron detector has been designed and constructed. This detector can achieve high detection efficiency. The detection range is 5 × 1000 mm^2^, and the thickness in the beam direction is 40 mm. An improved D-T neutron source with a rotating tritium target and ECR deuterium ion source [21,22] could operate for tens of hours at a yield of 1.0 × 10^11^ neutrons/s. Preliminary experiments of fast neutron scanning imaging and neutron tomography were carried out by this new system. The feasibility of this large-size line array fast neutron detector and new neutron source was verified. At the moment, the scanning time of a sample 200 mm in height is about 1.5 h, and the fast neutron tomography requires about 10 h.

## 2. Facility

### 2.1. System Overview

Our scanning method is similar to X-ray technology, the sample is lifted up and down by a bearing system, and the neutron source and detector are fixed, as shown in Figure 1. The data-acquiring system collects transmission projection images of each horizontal layer and the image splicing algorithm exports the holistic structure imaging results. Different from the original neutron imaging facility [14], the D-T neutron source and fast neutron collimator have both been upgraded. The whole system is shown in Figure 2. Additionally, traditional neutron tomography could be realized by rotating the sample in this system.

Table 1 shows the comparison between this system and the above-mentioned devices. It can be seen that on a relatively low neutron yield, this system can achieve the maximum detection range with considerable spatial resolution.

### 2.2. Neutron Source

A cascade accelerator supplies 14 MeV neutrons from the D-T fusion reaction [14]. Because deuterium beam power density depositing at the tritium target is very high, in the original system, each tritium target could work for a few hours at the highest intensity. In order to increase the lifetime of a single target and reduce the frequency of the target replacement, a new rotating target structure has been designed and manufactured, as shown in Figure 3a. The new tritium target area is larger than the original ones, which can rotate perpendicular to the deuterium beam direction at the rate of 200 cycles/min. In front of the rotating structure, a cross-analysis magnet is added to choose D+ particles from different kinds of deuterium particles. It could decrease useless particle energy deposition at the tritium target. A compact ECR deuterium ion source is also applied to replace the former high-frequency deuterium ion source. It could supply a more stable deuterium ion beam of 1 mA. Finally, the new D-T source could operate for over 40 h at the yield of 1.0 × 10^11^ neutrons/s, as shown in Figure 3b. The abscissa represents the time of neutron source emergence, and the ordinate represents the neutron yield.

### 2.3. Collimator

As shown in Figure 4, a special collimator has been designed and constructed. It adds a fan-shaped structure for line array scanning and retains cone structure for traditional area array imaging. With a hydraulic bearing platform, the whole collimator could be moved in the vertical direction to suit different imaging types. At a distance of 2 m from the source target, the field of neutrons from the cone structure is 200 × 200 mm^2^ and from the fan structure, 15 × 600 mm^2^.

A cylinder copper is used as a primary shielding layer to lower the neutron energy from 14 MeV to several MeVs, furthermore, a boron-polyethylene is used as secondary shielding to reduce neutron energy. Monte-Carlo simulation using Geant4 [23] with the cross-section library of ENDF/B VII [24] has been carried out to evaluate the collimator design. The results show that copper cylinder radius of 15 cm and boron-polyethylene thickness of 70 cm could reduce fast neutron intensity in the background area to less than 10% compared with the FOV area. The whole collimator is covered with 2 cm lead to reduce the gamma dose and 1 cm aluminum to protect inner structures. The assembling diagram and a picture of the collimator are shown in Figure 5, the whole collimator is a part of the cylinder, and the total weight is about 4.5 tons, which could be moved in the vertical direction by a hydraulic bearing platform.

### 2.4. Line Array Scintillator

The key detection module for fast neutron line array scanning is a polypropylene zinc sulfide scintillator embedded with wavelength-shifting fibers. Fiber diameter and arrangement need to be optimized. The scintillation material is used to detect fast neutrons, but the wave-shifting fiber cannot detect fast neutrons. When the diameter of the fiber and the thickness of the scintillator are constant, the smaller the center distance of the fibers is, the easier the neutron inducing photons can enter the wavelength-shifting optical fibers. It is good to improve the detection efficiency. However, at the same time, due to the increase of fiber spacing, the whole scintillation material decreases, which leads to the decline of the overall detection efficiency. Therefore, the center distance of fibers has an optimal value, which can balance the transmission ability of photons and the conversion ability of neutrons.

With the calculation method of [19], a neutron-inducing photon propagation model is established by the numerical simulation method. The simulation calculation is carried out for different fiber diameters, fiber center distance, arrangement modes, etc. The best fiber center distance is around 1.1 mm for 0.5 mm diameter fibers. With the same method, the result is around 0.8 mm for 0.3 mm diameter fibers. A larger or smaller center distance of fibers will cause the total detection efficiency decrease, as shown in Figure 6. When the CCD is 2048 × 2048 pixels and the scintillator length is about 1 m, about 0.5 mm for each pixel. To balance resolution and efficiency, fiber center distance is finally chosen to be 0.5 mm for 0.3 mm diameter fibers. The efficiency is about 60% of the best setting.

A line array scintillator based on polypropylene zinc sulfide embedded with wavelength-shifting fibers was constructed. Ten row fibers (2000 for each) are embedded in polypropylene zinc sulfide. A fixed structure is constructed to fasten fibers one by one. After all fibers in one row are fastened, then polypropylene zinc sulfide is deposited and solidified. Repeating this process could finally get the whole scintillator. The length of all fibers is 40 mm. The whole scintillator is protected by epoxy-polyethylene and covered by black paper. The construction process and the scintillator are shown in Figure 7.

### 2.5. Imaging System

A digital imaging system is composed of a CCD camera, a line array scintillator, a line array reflector, and a Nikon lens. One scientific liquid nitrogen-cooled CCD camera with a chip size of 27.6 × 27.6 mm^2^ and 2048 × 2048 pixels is used to record the light picture. The design of the light path is shown in Figure 8. A Nikon lens of 50 mm focus distance cooperates with the CCD camera. The light FOV is about 1000 × 1000 mm^2^, while the total distance between the scintillator and CCD is about 1900 mm. A reflector is set between the scintillator and the CCD camera to reflect light for 90 degrees. It could avoid neutron radiating to the CCD chip directly.

**Figure 8 materials-15-04946-f008:**
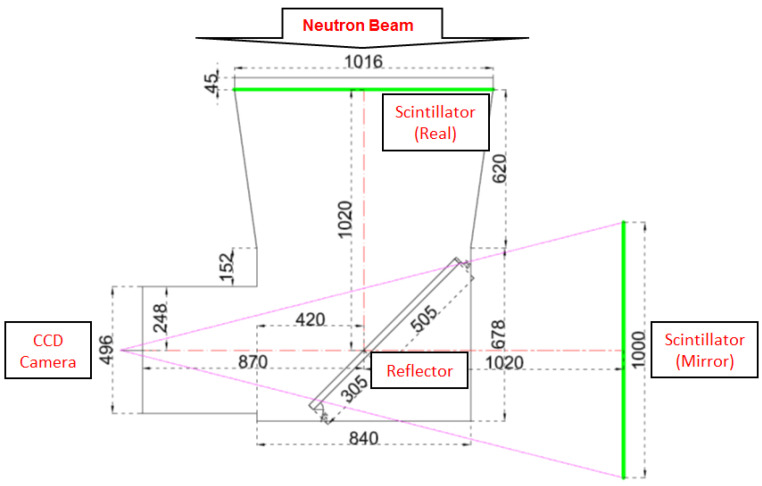
Overhead view of the light path. The green lines indicate the real image and the mirror image.The imaging system is covered by a dark case, which is composed of aluminum, boron polyethylene, and lead materials. Its inner height is 300 mm. The sketch map is shown in Figure 9.

**Figure 9 materials-15-04946-f009:**
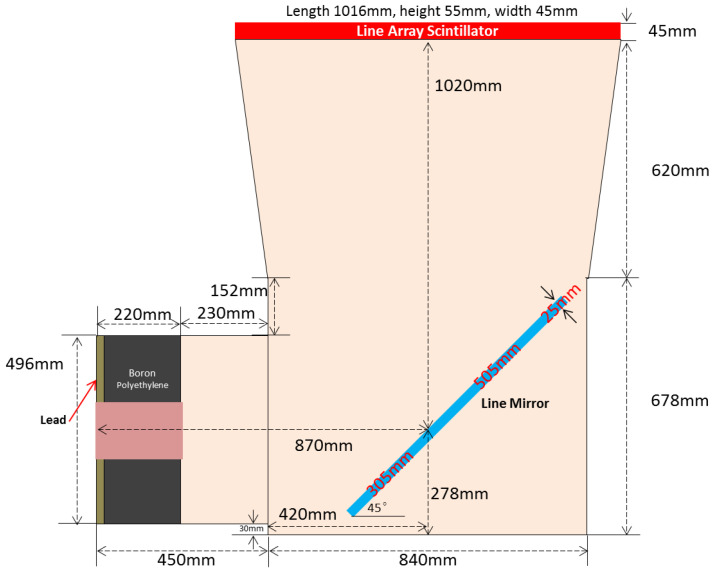
Overhead view of the dark case construction.

## 3. Imaging Results

### 3.1. Scanning Imaging

Preliminary experiments on this system have been carried out. The experimental process of scanning imaging is shown in Figure 10. With the sample moved up and down, the white images and different horizontal layer images are collected. The total scanning imaging results are finally merged after one-layer pro-dealing. 

The scanning imaging sample includes a motor and a stainless-steel cube (with different slits), as shown in Figure 11. The size of the stainless-steel cube is 50 mm (height) × 50 mm (thickness) × 110 mm (length), the depth of slits is 20 mm, and the width of slits ranges from 0.1 to 1 mm, respectively. 

In the experiment, the motor was stacked on the stainless-steel cube, and the slits were close to the scintillator. The different horizontal layer images were obtained by lifting the sample at 3 mm equal intervals in the vertical direction. Every horizontal layer collected five images, each exposed for 30 s, when neutron yield was 1 × 10^11^/s. Seventy horizontal layers were collected. By splicing the images of multiple horizontal layers, the scanning image was finally obtained, as shown in the lower left of Figure 11. 

The internal structure of the motor is clear, and multiple slits in the stainless-steel cube are visible to the naked eye. Through the gray value analysis, as shown in the lower right of Figure 11, the slits wider than 0.5 mm can be clearly distinguished. The results indicate that the scanning method based on line array fast neutron detector and D-T neutron source is feasible.

### 3.2. Fast Neutron Tomography

Based on this system, we have carried out fast neutron tomography experiments too. The test sample is made of polytetrafluoroethylene and copper, as shown in Figure 12. The ring sample’s external diameter is 200 mm, and polytetrafluoroethylene inside with a 5 mm copper shell. Holes are pierced in the light material, with diameters ranging from 10 to 2 mm, respectively. The cube sample’s external size is 60 × 110 mm^2^, and polytetrafluoroethylene inside with a 5 mm copper shell. Slits are grooved in inner light material, with widths ranging from 1 to 0.1 mm, respectively. 

In the experiment, with the sample rotated once every 1°, 1444 projection images (four images at each position) were obtained at 360°. Each image was exposed for 20 s when neutron yield was 1 × 10^11^ neutrons/s. With the same reconstruction method of ART from [13,25], the reconstructed image was obtained, as shown in Figure 12. The results demonstrate that 3 mm holes, or even larger, and 0.6 mm slits, or even wider, could be distinguished clearly.

## 4. Conclusions

A novel scanning system based on line array scintillator and D-T neutron source was designed and constructed. With a rotating tritium target and an ECR deuterium ion source, the long lifetime D-T neutron source was accomplished. A multi-use collimator with cone shape and fan shape was integrated into one structure. A line array scintillator based on a polypropylene zinc sulfide scintillator embedded with wavelength-shifting fibers was designed and constructed, which could realize a detection range of 5 × 1000 mm. Both fast neutron scanning imaging and fast neutron tomography were carried out with this new system. A slit of 0.5 mm width could be detected by scanning imaging, and a slit of 0.6 mm width could be detected by tomography. In the future, another line-typed fast neutron detector based on an array of plastic scintillators coupled with an array of SiPM readout systems will be applied in this system.

## Figures and Tables

**Figure 1 materials-15-04946-f001:**
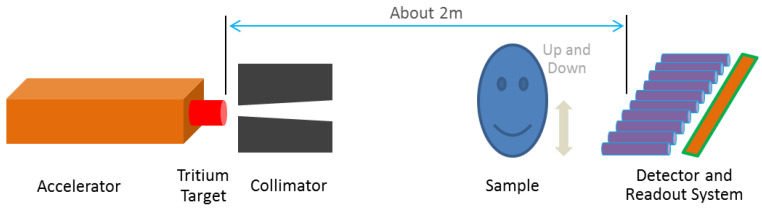
Sketch map of the scanning method by fast neutrons.

**Figure 2 materials-15-04946-f002:**
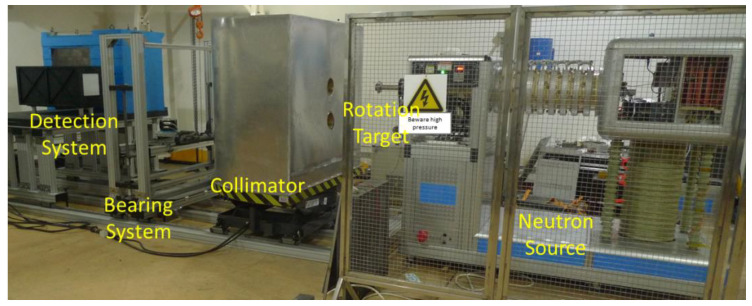
Picture of the fast neutron scanning imaging system.

**Figure 3 materials-15-04946-f003:**
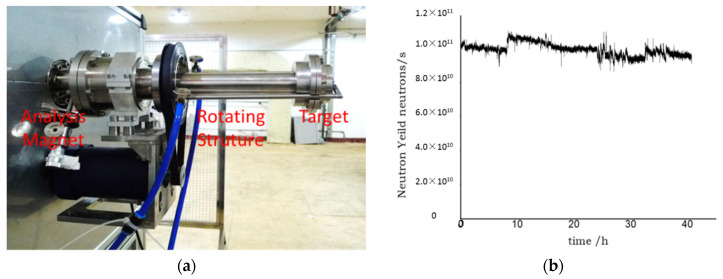
Picture of neutron source: (**a**) Rotating tritium target, (**b**) lifetime of the D-T neutron source.

**Figure 4 materials-15-04946-f004:**
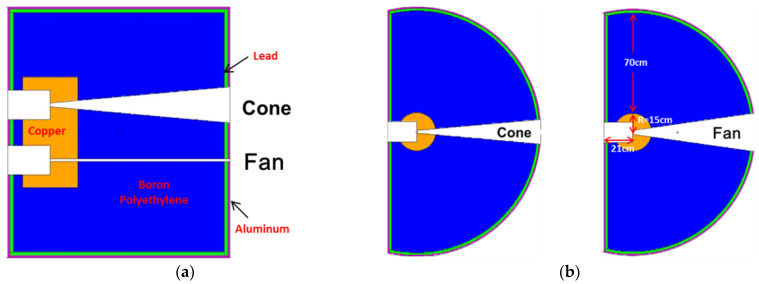
Diagram of collimator: (**a**) Cut-open view in the vertical direction, (**b**) cut-open view in the horizontal direction.

**Figure 5 materials-15-04946-f005:**
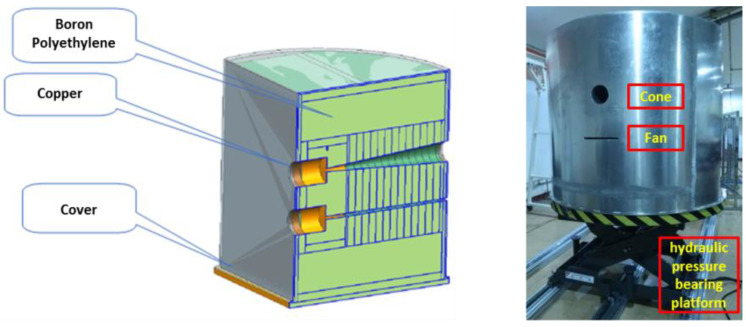
Assembling diagram and picture of the collimator.

**Figure 6 materials-15-04946-f006:**
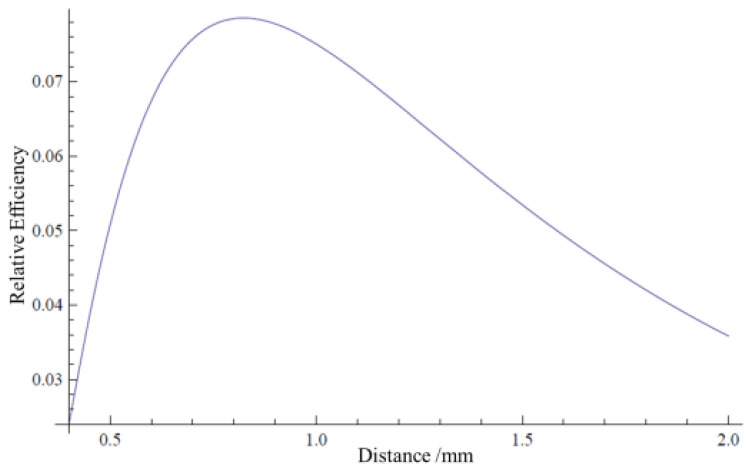
Relative detecting efficiency of different fiber center distances.

**Figure 7 materials-15-04946-f007:**
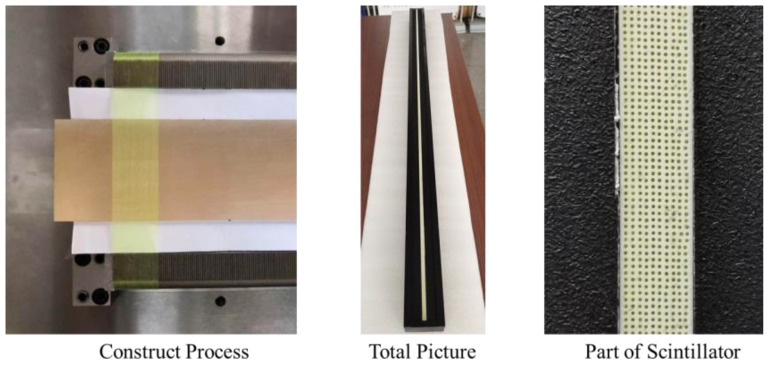
Pictures of the line array scintillator.

**Figure 10 materials-15-04946-f010:**
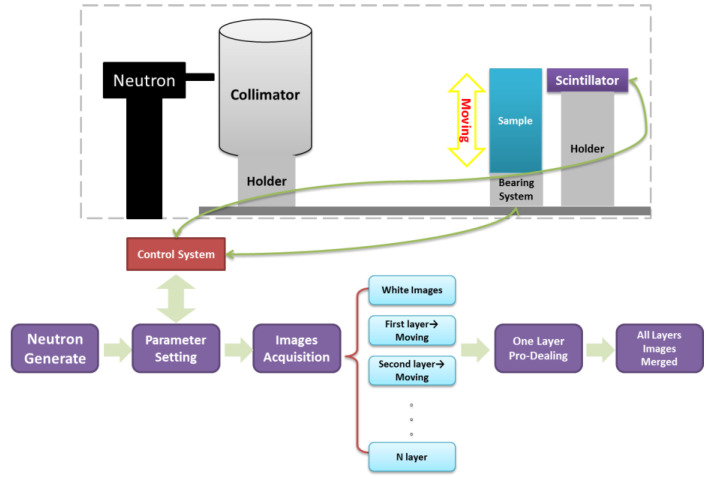
Experimental process of scanning imaging.

**Figure 11 materials-15-04946-f011:**
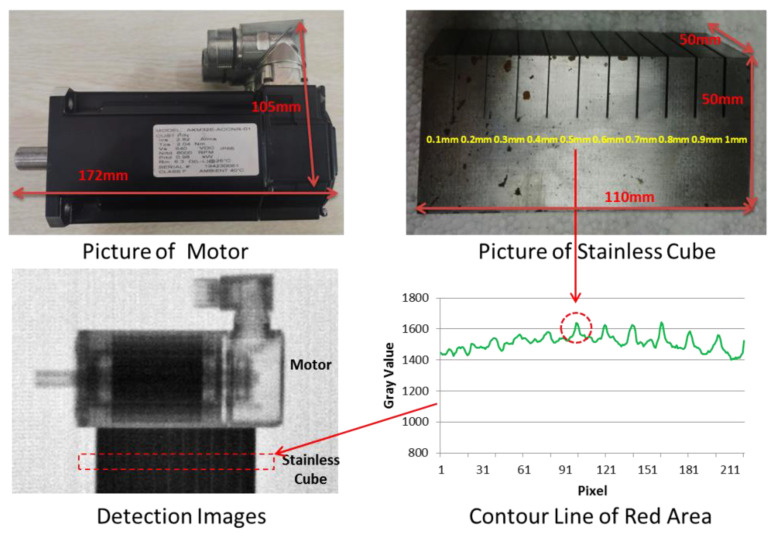
Preliminary scanning imaging results.

**Figure 12 materials-15-04946-f012:**
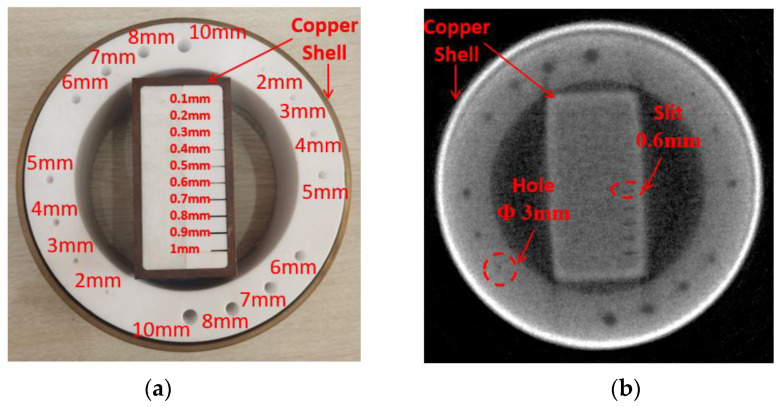
Sample and reconstructed image of fast tomography. (**a**) Sample of fast neutron tomography; (**b**) Reconstructed slice image.

**Table 1 materials-15-04946-t001:** The comparison between this system and other systems on characteristic parameters.

Facility Name	Accelerator Type	Neutron Flux (n/cm^2^/s)	Field of View (mm)	Spatial Resolution (mm)	Tomography
This Facility	Cascade	~2 × 10^5^	1000	0.5	yes
LENS [8]	Linac	~1 × 10^6^	200	0.2	yes
PTB Accelerator Facility [9]	Linac	2–4 × 10^5^	330	~0.5	no
Necsa RFQ Facility [10]	Linac	~1.5 × 10^4^	~200	~0.4	no

## Data Availability

Not applicable.

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
