# Peer review of "A Novel NDT Scanning System Based on Line Array Fast Neutron Detector and D-T Neutron Source"

_materials, 2022, doi:10.3390/ma15144946_

Round 1

Reviewer 1 Report

> This paper is about D-T neutron based non-destructive imaging.

>
> Neutron imaging based on the combination of D-T generator and neutron
> detector consisting of scintillating fiber, a reflector and a CCD camera
> is relatively new and valuable partially because the whole system
> required expensive, large size and heavy devices installed in a heavily
> shielding room. The overall construction including detailed instruments
> such as a rotating target material and multiple shielding have their own
> values.
>
> The performance of the system is evaluated using a standardized phantom
> and the reconstructed neutron images represent the shape of phantom and
> the resolution of the imaging system. The conclusions are consistent
> with the evidence and arguments presented.
>
> The overall contents are neutron emission and imaging which could be
> somewhat different from the name of journal ‘material’. However, the
> results are interesting and beneficial to readers in the field of
> neutron applications. The paper is mostly well written and the text is
> clear and easy to read.
>
> In conclusion, this paper contents show originality and practicality,
> and hence, the paper is good to be published with minor revisions.
>
> Minor:
>
> Fig. 6: If the distance between fibers is smaller, the efficiency should
> be higher. In the Figure, however, a maximum distance exists in the
> middle of distance. Why the efficiency decreased below 0.8 mm?
>
> Line 32: holes’ flaws, Line 114: fiber’s centers, Line 120: fiber’s
> center, Line 123: fiber's length, Line 145 and 145: layers’ images ->
> English needs to be improved.

Reviewer 2 Report

The paper presents first results from a new fast neutron radiography/tomography system based on a D-T neutron source and a detection array. The system is the evolution of a previous set-up which had limited performance. 

The first results of radiography and tomography measurements presented here give an idea of the performance to be expected from the new system. A thourough quantitative assessment of the performance in terms of sensitivity, s/n etc is outside the scope of this paper.

The main message is that the system exists and is working. Since there are few, if any, such systems worldwide, this is a message that is worth a publication.

I therefore recommend the paper be published after the authors have adequately 

described the broader context: what is unique about this facility? Are there other fast neutron radiography systems with similar or better performance in operation worldwide? How does DT neutron radiography/tomography compare with thermal/cold neutron tomography; i.e. is thermal/cold neutron radiography/tomography of the two samples shown in the paper possible?

The latter question should be addressed with reference to neutron tomography systems available to users at large scale facilities such as PSI (Villigen, CH), ISIS (Didcot, UK), ORNL (Oak Ridge, USA). 

Reviewer 3 Report

The authors present a new NDT scanning system using a compact D-T neutron source. Scanning range and resolution are investigated. 

I have several issues with the manuscript, especially the presentation, in its current form that I would like to see addressed before I can recommend publication. 

1. Introduction: Here I expect a general statement of the scientific problem, followed by the current status of the research and emphasis how the work presented in the paper is going to build and improve upon this. Although some of this might have been written by the authors, the overall presentation is frustratingly murky and unclear. This might be due to the overuse of passive voice. Please state clearly what you want to improve and what your achievement is. 

2. Figures 3.b, 6, 11: The plots of data that are presented are not explained and barely referenced throughout the text. How were these figures obtained and what do I see there? What is the conclusion drawn from these? That needs to be explained in detail in the text. 

3. I recommend seeking the help of a professional language editing service to improve the overall readability of the manuscript. 
